# Robust Kernel Density Estimation by Scaling and Projection in Hilbert Space

**Robert A. Vandermeulen**
Department of EECS
University of Michigan
Ann Arbor, MI 48109
rvdm@umich.edu

**Clayton D. Scott**
Deparment of EECS
Univeristy of Michigan
Ann Arbor, MI 48109
clayscot@umich.edu

## Abstract

While robust parameter estimation has been well studied in parametric density estimation, there has been little investigation into robust density estimation in the nonparametric setting. We present a robust version of the popular kernel density estimator (KDE). As with other estimators, a robust version of the KDE is useful since sample contamination is a common issue with datasets. What "robustness" means for a nonparametric density estimate is not straightforward and is a topic we explore in this paper. To construct a robust KDE we scale the traditional KDE and project it to its nearest weighted KDE in the $L^2$ norm. This yields a scaled and projected KDE (SPKDE). Because the squared $L^2$ norm penalizes point-wise errors superlinearly this causes the weighted KDE to allocate more weight to high density regions. We demonstrate the robustness of the SPKDE with numerical experiments and a consistency result which shows that asymptotically the SPKDE recovers the uncontaminated density under sufficient conditions on the contamination.

## 1 Introduction

The estimation of a probability density function (pdf) from a random sample is a ubiquitous problem in statistics. Methods for density estimation can be divided into parametric and nonparametric, depending on whether parametric models are appropriate. Nonparametric density estimators (NDEs) offer the advantage of working under more general assumptions, but they also have disadvantages with respect to their parametric counterparts. One of these disadvantages is the apparent difficulty in making NDEs robust, which is desirable when the data follow not the density of interest, but rather a contaminated version thereof. In this work we propose a robust version of the KDE, which serves as the workhorse among NDEs [11, 10].

We consider the situation where most observations come from a target density $f_{tar}$ but some observations are drawn from a contaminating density $f_{con}$, so our observed samples come from the density $f_{obs} = (1 - \varepsilon) f_{tar} + \varepsilon f_{con}$. It is not known which component a given observation comes from. When considering this scenario in the infinite sample setting we would like to construct some transform that, when applied to $f_{obs}$, yields $f_{tar}$. We introduce a new formalism to describe transformations that "decontaminate" $f_{obs}$ under sufficient conditions on $f_{tar}$ and $f_{con}$. We focus on a specific nonparametric condition on $f_{tar}$ and $f_{con}$ that reflects the intuition that the contamination manifests in low density regions of $f_{tar}$. In the finite sample setting, we seek a NDE that converges to $f_{tar}$ asymptotically. Thus, we construct a weighted KDE where the kernel weights are lower in low density regions and higher in high density regions. To do this we multiply the standard KDE by a real value greater than one (scale) and then find the closest pdf to the scaled KDE in the $L^2$ norm (project), resulting in a scaled and projected kernel density estimator (SPKDE). Because the squared $L^2$ norm penalizes point-wise differences between functions quadratically, this causes the

SPKDE to draw weight from the low density areas of the KDE and move it to high density areas to get a more uniform difference to the scaled KDE. The asymptotic limit of the SPKDE is a scaled and shifted version of $f_{obs}$. Given our proposed sufficient conditions on $f_{tar}$ and $f_{con}$, the SPKDE can asymptotically recover $f_{tar}$.

A different construction for a robust kernel density estimator, the aptly named "robust kernel density estimator" (RKDE), was developed by Kim & Scott [6]. In that paper the RKDE was analytically and experimentally shown to be robust, but no consistency result was presented. Vandermeulen & Scott [15] proved that a certain version of the RKDE converges to $f_{obs}$. To our knowledge the convergence of the SPKDE to a transformed version of $f_{obs}$, which is equal to $f_{tar}$ under sufficient conditions on $f_{tar}$ and $f_{con}$, is the first result of its type.

In this paper we present a new formalism for nonparametric density estimation, necessary and sufficient conditions for decontamination, the construction of the SPKDE, and a proof of consistency. We also include experimental results applying the algorithm to benchmark datasets with comparisons to the RKDE, traditional KDE, and an alternative robust KDE implementation. Many of our results and proof techniques are novel in KDE literature. Proofs are contained in the supplemental material.

## 2 Nonparametric Contamination Models and Decontamination Procedures for Density Estimation

What assumptions are necessary and sufficient on a target and contaminating density in order to theoretically recover the target density is a question that, to the best of our knowledge, is completely unexplored in a nonparametric setting. We will approach this problem in the infinite sample setting, where we know $f_{obs} = (1 - \varepsilon)f_{tar} + \varepsilon f_{con}$ and $\varepsilon$, but do not know $f_{tar}$ or $f_{con}$. To this end we introduce a new formalism. Let $\mathcal{D}$ be the set of all pdfs on $\mathbb{R}^d$. We use the term *contamination model* to refer to any subset $\mathcal{V} \subset \mathcal{D} \times \mathcal{D}$, i.e. a set of pairs $(f_{tar}, f_{con})$. Let $R_\varepsilon : \mathcal{D} \to \mathcal{D}$ be a set of transformations on $\mathcal{D}$ indexed by $\varepsilon \in [0, 1)$. We say that $R_\varepsilon$ *decontaminates* $\mathcal{V}$ if for all $(f_{tar}, f_{con}) \in \mathcal{V}$ and $\varepsilon \in [0, 1)$ we have $R_\varepsilon((1 - \varepsilon)f_{tar} + \varepsilon f_{con}) = f_{tar}$.

One may wonder whether there exists some set of contaminating densities, $\mathcal{D}_{con}$, and a transformation, $R_\varepsilon$, such that $R_\varepsilon$ decontaminates $\mathcal{D} \times \mathcal{D}_{con}$. In other words, does there exist some set of contaminating densities for which we can recover any target density? It turns out this is impossible if $\mathcal{D}_{con}$ contains at least two elements.

**Proposition 1.** *Let $\mathcal{D}_{con} \subset \mathcal{D}$ contain at least two elements. There does not exist any transformation $R_\varepsilon$ which decontaminates $\mathcal{D} \times \mathcal{D}_{con}$.*

*Proof.* Let $f \in \mathcal{D}$ and $g, g' \in \mathcal{D}_{con}$ such that $g \neq g'$. Let $\varepsilon \in (0, \frac{1}{2})$. Clearly $f_{tar} \triangleq \frac{f(1-2\varepsilon)+g\varepsilon}{1-\varepsilon}$ and $f'_{tar} \triangleq \frac{f(1-2\varepsilon)+\varepsilon g'}{1-\varepsilon}$ are both elements of $\mathcal{D}$. Note that

$$(1 - \varepsilon)f_{tar} + \varepsilon g' = (1 - \varepsilon)f'_{tar} + \varepsilon g.$$

In order for $R_\varepsilon$ to decontaminate $\mathcal{D}$ with respect to $\mathcal{D}_{con}$, we need $R_\varepsilon\left((1 - \varepsilon)f_{tar} + \varepsilon g'\right) = f_{tar}$ and $R_\varepsilon\left((1 - \varepsilon)f'_{tar} + \varepsilon g\right) = f'_{tar}$, which is impossible since $f_{tar} \neq f'_{tar}$. $\square$

This proposition imposes significant limitations on what contamination models can be decontaminated. For example, suppose we know that $f_{con}$ is Gaussian with known covariance matrix and unknown mean. Proposition 1 says we cannot design $R_\varepsilon$ so that it can decontaminate $(1-\varepsilon)f_{tar}+\varepsilon f_{con}$ for all $f_{tar} \in \mathcal{D}$. In other words, it is impossible to design an algorithm capable of removing Gaussian contamination (for example) from arbitrary target densities. Furthermore, if $R_\varepsilon$ decontaminates $\mathcal{V}$ and $\mathcal{V}$ is fully nonparametric (i.e. for all $f \in \mathcal{D}$ there exists some $f' \in \mathcal{D}$ such that $(f, f') \in \mathcal{V}$) then for each $(f_{tar}, f_{con})$ pair, $f_{con}$ must satisfy some properties which depend on $f_{tar}$.

### 2.1 Proposed Contamination Model

For a function $f : \mathbb{R}^d \to \mathbb{R}$ let $\mathrm{supp}(f)$ denote the support of $f$. We introduce the following contamination assumption:

**Assumption A.** *For the pair* $(f_{tar}, f_{con})$, *there exists* $u$ *such that* $f_{con}(x) = u$ *for almost all (in the Lebesgue sense)* $x \in \mathrm{supp}(f_{tar})$ *and* $f_{con}(x') \leq u$ *for almost all* $x' \notin \mathrm{supp}(f_{tar})$.

See Figure 1 for an example of a density satisfying this assumption. Because $f_{con}$ must be uniform over the support of $f_{tar}$ a consequence of Assumption A is that $\mathrm{supp}(f_{tar})$ has finite Lebesgue measure. Let $\mathcal{V}_A$ be the contamination model containing all pairs of densities which satisfy Assumption A. Note that $\bigcup_{(f_{tar}, f_{con}) \in \mathcal{V}_A} f_{tar}$ is exactly all densities whose support has finite Lebesgue measure, which includes all densities with compact support.

The uniformity assumption on $f_{con}$ is a common "noninformative" assumption on the contamination. Furthermore, this assumption is supported by connections to one-class classification. In that problem, only one class (corresponding to our $f_{tar}$) is observed for training, but the testing data is drawn from $f_{obs}$ and must be classified. The dominant paradigm for nonparametric one-class classification is to estimate a level set of $f_{tar}$ from the one observed training class [14, 7, 13, 16, 12, 9], and classify test data according to that level set. Yet level sets only yield optimal classifiers (i.e. likelihood ratio tests) under the uniformity assumption on $f_{con}$, so that these methods are implicitly adopting this assumption. Furthermore, a uniform contamination prior has been shown to optimize the worst-case detection rate among all choices for the unknown contamination density [5]. Finally, our experiments demonstrate that the SPKDE works well in practice, even when Assumption A is significantly violated.

## 2.2 Decontamination Procedure

Under Assumption A $f_{tar}$ is present in $f_{obs}$ and its shape is left unmodified (up to a multiplicative factor) by $f_{con}$. To recover $f_{tar}$ it is necessary to first scale $f_{obs}$ by $\beta = \frac{1}{1-\varepsilon}$ yielding

$$\frac{1}{1-\varepsilon}\left((1-\varepsilon)f_{tar} + \varepsilon f_{con}\right) = f_{tar} + \frac{\varepsilon}{1-\varepsilon}f_{con}. \tag{1}$$

After scaling we would like to slice off $\frac{\varepsilon}{1-\varepsilon}f_{con}$ from the bottom of $f_{tar} + \frac{\varepsilon}{1-\varepsilon}f_{con}$. This transform is achieved by

$$\max\left\{0, f_{tar} + \frac{\varepsilon}{1-\varepsilon}f_{con} - \alpha\right\}, \tag{2}$$

where $\alpha$ is set such that 2 is a pdf (which in this case is achieved with $\alpha = r\frac{\varepsilon}{1-\varepsilon}$). We will now show that this transform is well defined in a general sense. Let $f$ be a pdf and let

$$g_{\beta,\alpha} = \max\left\{0, \beta f(\cdot) - \alpha\right\}$$

where the max is defined pointwise. The following lemma shows that it is possible to slice off the bottom of any scaled pdf to get a transformed pdf and that the transformed pdf is unique.

**Lemma 1.** *For fixed* $\beta > 1$ *there exists a unique* $\alpha' > 0$ *such that* $\|g_{\beta,\alpha'}\|_{L^1} = 1$.

Figure 2 demonstrates this transformation applied to a pdf. We define the following transform $R_\varepsilon^A : \mathcal{D} \to \mathcal{D}$ where $R_\varepsilon^A(f) = \max\left\{\frac{1}{1-\varepsilon}f(\cdot) - \alpha, 0\right\}$ where $\alpha$ is such that $R_\varepsilon^A(f)$ is a pdf.

**Proposition 2.** $R_\varepsilon^A$ *decontaminates* $\mathcal{V}_A$.

The proof of this proposition is an intermediate step for the proof for Theorem 2. For any two subsets of $\mathcal{V}, \mathcal{V}' \subset \mathcal{D} \times \mathcal{D}$, $R_\varepsilon$ decontaminates $\mathcal{V}$ and $\mathcal{V}'$ iff $R_\varepsilon$ decontaminates $\mathcal{V} \bigcup \mathcal{V}'$. Because of this, every decontaminating transform has a maximal set which it can decontaminate. Assumption A is both sufficient and necessary for decontamination by $R_\varepsilon^A$, i.e. the set $\mathcal{V}_A$ is maximal.

**Proposition 3.** *Let* $\{(q, q')\} \in \mathcal{D} \times \mathcal{D}$ *and* $(q, q') \notin \mathcal{V}_A$. $R_\varepsilon^A$ *cannot decontaminate* $\{(q, q')\}$.

The proof of this proposition is in the supplementary material.

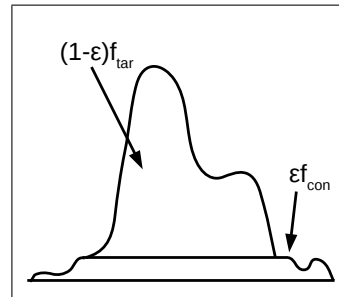

Figure 1: Density with contamination satisfying Assumption A

## 2.3 Other Possible Contamination Models

The model described previously is just one of many possible models. An obvious approach to robust kernel density estimation is to use an anomaly detection algorithm and construct the KDE using only non-anomalous samples. We will investigate this model under a couple of anomaly detection schemes and describe their properties.

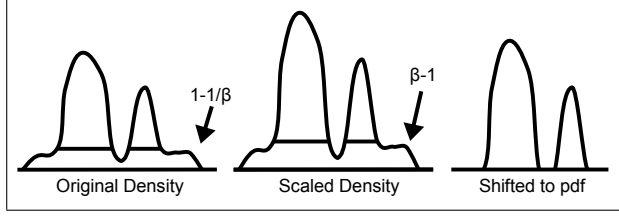

Figure 2: Infinite sample SPKDE transform. Arrows indicate the area under the line.

One of the most common methods for anomaly detection is the level set method. For a probability measure $\mu$ this method attempts to find the set $S$ with smallest Lebesgue measure such that $\mu(S)$ is above some threshold, $t$, and declares samples outside of that set as being anomalous. For a density $f$ this is equivalent to finding $\lambda$ such that $\int_{\{x|f(x)\geq\lambda\}} f(y)dy = t$ and declaring samples were $f(X) < \lambda$ as being anomalous. Let $X_1, \ldots, X_n$ be iid samples from $f_{obs}$. Using the level set method for a robust KDE, we would construct a density $\widehat{f}_{obs}$ which is an estimate of $f_{obs}$. Next we would select some threshold $\lambda > 0$ and declare a sample, $X_i$, as being anomalous if $\widehat{f}_{obs}(X_i) < \lambda$. Finally we would construct a KDE using the non-anomalous samples. Let $\chi_{\{\cdot\}}$ be the indicator function. Applying this method in the infinite sample situation, i.e. $\widehat{f}_{obs} = f_{obs}$, would cause our non-anomalous samples to come from the density $p(x) = \frac{f_{obs}(x)\chi_{\{f_{obs}(x)>\lambda\}}}{\tau}$ where $\tau = \int \chi_{\{f(y)>\lambda\}} f(y)dy$. See Figure 3. Perfect recovery of $f_{tar}$ using this method requires $\varepsilon f_{con}(x) \leq f_{tar}(x)(1-\varepsilon)$ for all $x$ and that $f_{con}$ and $f_{tar}$ have disjoint supports. The first assumption means that this density estimator can only recover $f_{tar}$ if it has a drop off on the boundary of its support, whereas Assumption A only requires that $f_{tar}$ have finite support. See the last diagram in Figure 3. Although these assumptions may be reasonable in certain situations, we find them less palatable than Assumption A. We also evaluate this approach experimentally later and find that it performs poorly.

Another approach based on anomaly detection would be to find the connected components of $f_{obs}$ and declare those that are, in some sense, small as being anomalous. A "small" connected component may be one that integrates to a small value, or which has a small mode. Unfortunately this approach also assumes that $f_{tar}$ and $f_{con}$ have disjoint supports. There are also computational issues with this anomaly detection scheme; finding connected components, finding modes, and numerical integration are computationally difficult.

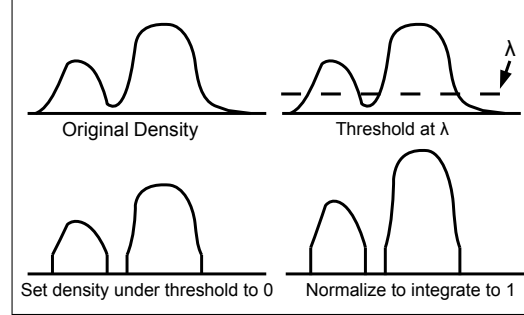

Figure 3: Infinite sample version of the level set rejection KDE

To some degree, $R_\varepsilon^A$ actually achieves the objectives of the previous two robust KDEs. For the first model, the $R_\varepsilon^A$ does indeed set those regions of the pdf that are below some threshold to zero. For the second, if the magnitude of the level at which we choose to slice off the bottom of the contaminated density is larger than the mode of the anomalous component then the anomalous component will be eliminated.

## 3 Scaled Projection Kernel Density Estimator

Here we consider approximating $R_\varepsilon^A$ in a finite sample situation. Let $f \in L^2(\mathbb{R}^d)$ be a pdf and $X_1, \ldots, X_n$ be iid samples from $f$. Let $k_\sigma(x, x')$ be a radial smoothing kernel with bandwidth $\sigma$ such that $k_\sigma(x, x') = \sigma^{-d} q(\|x - x'\|_2 / \sigma)$, where $q(\|\cdot\|_2) \in L^2(\mathbb{R}^d)$ and is a pdf. The classic kernel density estimator is:

$$\bar{f}_\sigma^n := \frac{1}{n} \sum_1^n k_\sigma(\cdot, X_i).$$

In practice $\varepsilon$ is usually not known and Assumption A is violated. Because of this we will scale our density by $\beta > 1$ rather than $\frac{1}{1-\varepsilon}$. For a density $f$ define

$$Q_\beta(f) \triangleq \max\left\{\beta f\left(\cdot\right) - \alpha, 0\right\},$$

where $\alpha = \alpha(\beta)$ is set such that the RHS is a pdf. $\beta$ can be used to tune robustness with larger $\beta$ corresponding to more robustness (setting $\beta$ to 1 in all the following transforms simply yields the KDE). Given a KDE we would ideally like to apply $Q_\beta$ directly and search over $\alpha$ until $\max\left\{\beta \bar{f}_\sigma^n\left(\cdot\right) - \alpha, 0\right\}$ integrates to 1. Such an estimate requires multidimensional numerical integration and is not computationally tractable. The SPKDE is an alternative approach that always yields a density and manifests the transformed density in its asymptotic limit.

We now introduce the construction of the SPKDE. Let $\mathcal{D}_\sigma^n$ be the convex hull of $k_\sigma\left(\cdot, X_1\right), \ldots, k_\sigma\left(\cdot, X_n\right)$ (the space of weighted kernel density estimators). The SPKDE is defined as

$$f_{\sigma,\beta}^n := \arg\min_{g \in \mathcal{D}_\sigma^n} \left\|\beta \bar{f}_\sigma^n - g\right\|_{L^2},$$

which is guaranteed to have a unique minimizer since $\mathcal{D}_\sigma^n$ is closed and convex and we are projecting in a Hilbert space ([1] Theorem 3.14). If we represent $f_{\sigma,\beta}^n$ in the form

$$f_{\sigma,\beta}^n = \sum_1^n a_i k_\sigma\left(\cdot, X_i\right),$$

then the minimization problem is a quadratic program over the vector $a = [a_1, \ldots, a_n]^T$, with $a$ restricted to the probabilistic simplex, $\Delta^n$. Let $G$ be the Gram matrix of $k_\sigma\left(\cdot, X_1\right), \ldots, k_\sigma\left(\cdot, X_n\right)$, that is

$$
\begin{aligned}
G_{ij} &= \left\langle k_\sigma\left(\cdot, X_i\right), k_\sigma\left(\cdot, X_j\right)\right\rangle_{L^2} \\
&= \int k_\sigma\left(x, X_i\right) k_\sigma\left(x, X_j\right) dx.
\end{aligned}
$$

Let $\mathbf{1}$ be the ones vector and $b = G\mathbf{1}\frac{\beta}{n}$, then the quadratic program is

$$\min_{a \in \Delta^n} a^T G a - 2b^T a.$$

Since $G$ is a Gram matrix, and therefore positive-semidefinite, this quadratic program is convex. Furthermore, the integral defining $G_{ij}$ can be computed in closed form for many kernels of interest. For example for the Gaussian kernel

$$k_\sigma\left(x, x'\right) = \left(2\pi\sigma^2\right)^{-\frac{d}{2}} \exp\left(\frac{-\left\|x - x'\right\|^2}{2\sigma^2}\right) \implies G_{ij} = k_{\sqrt{2}\sigma}(X_i, X_j),$$

and for the Cauchy kernel [2]

$$k_\sigma\left(x, x'\right) = \frac{\Gamma\left(\frac{1+d}{2}\right)}{\pi^{(d+1)/2} \cdot \sigma^d} \left(1 + \frac{\left\|x - x'\right\|^2}{\sigma^2}\right)^{-\frac{1+d}{2}} \implies G_{ij} = k_{2\sigma}(X_i, X_j).$$

We now present some results on the asymptotic behavior of the SPKDE. Let $\mathcal{D}$ be the set of all pdfs in $L^2\left(\mathbb{R}^d\right)$. The infinite sample version of the SPKDE is

$$f_\beta' = \arg\min_{h \in \mathcal{D}} \|\beta f - h\|_{L^2}^2.$$

It is worth noting that projection operators in Hilbert space, like the one above, are known to be well defined if the convex set we are projecting onto is closed and convex. $\mathcal{D}$ is not closed in $L^2\left(\mathbb{R}^d\right)$, but this turns out not to be an issue because of the form of $\beta f$. For details see the proof of Lemma 2 in the supplemental material.

**Lemma 2.** $f_\beta' = \max\left\{\beta f\left(\cdot\right) - \alpha, 0\right\}$ *where $\alpha$ is set such that* $\max\left\{\beta f\left(\cdot\right) - \alpha, 0\right\}$ *is a pdf.*

Given the same rate on bandwidth necessary for consistency of the traditional KDE, the SPKDE converges to its infinite sample version in its asymptotic limit.

**Theorem 1.** *Let $f \in L^2\left(\mathbb{R}^d\right)$. If $n \to \infty$ and $\sigma \to 0$ with $n\sigma^d \to \infty$ then $\left\|f_{\sigma,\beta}^n - f_\beta'\right\|_{L^2} \xrightarrow{p} 0$.*

Because $f_{\sigma,\beta}^n$ is a sequence of pdfs and $f_\beta' \in L^2\left(\mathbb{R}^d\right)$, it is possible to show $L^2$ convergence implies $L^1$ convergence.

**Corollary 1.** *Given the conditions in the previous theorem statement, $\left\|f_{\sigma,\beta}^n - f_\beta'\right\|_{L^1} \xrightarrow{p} 0$.*

To summarize, the SPKDE converges to a transformed version of $f$. In the next section we will show that under Assumption A and with $\beta = \frac{1}{1-\varepsilon}$, the SPKDE converges to $f_{tar}$.

### 3.1 SPKDE Decontamination

Let $f_{tar} \in L^2\left(\mathbb{R}^d\right)$ be a pdf having support with finite Lebesgue measure and let $f_{tar}$ and $f_{con}$ satisfy Assumption A. Let $X_1, X_2, \ldots, X_n$ be iid samples from $f_{obs} = (1-\varepsilon)f_{tar} + \varepsilon f_{con}$ with $\varepsilon \in [0,1)$. Finally let $f_{\sigma,\beta}^n$ be the SPKDE constructed from $X_1, \ldots, X_n$, having bandwidth $\sigma$ and robustness parameter $\beta$. We have

**Theorem 2.** *Let $\beta = \frac{1}{1-\varepsilon}$. If $n \to \infty$ and $\sigma \to 0$ with $n\sigma^d \to \infty$ then $\left\|f_{\sigma,\beta}^n - f_{tar}\right\|_{L_1} \xrightarrow{p} 0$.*

To our knowledge this result is the first of its kind, wherein a nonparametric density estimator is able to asymptotically recover the underlying density in the presence of contaminated data.

## 4 Experiments

For all of the experiments optimization was performed using projected gradient descent. The projection onto the probabilistic simplex was done using the algorithm developed in [4] (which was actually originally discovered a few decades ago [3, 8]).

### 4.1 Synthetic Data

To show that the SPKDE's theoretical properties are manifested in practice we conducted an idealized experiment where the contamination is uniform and the contamination proportion is known. Figure 4 exhibits the ability of the SPKDE to compensate for uniform noise. Samples for the density estimator came from a mixture of the "Target" density with a uniform contamination on $[-2, 2]$, sampling from the contamination with probability $\varepsilon = 0.2$. This experiment used 500 samples and the robustness parameter $\beta$ was set to $\frac{1}{1-\varepsilon} = \frac{5}{4}$ (the value for perfect asymptotic decontamination).

The SPKDE performs well in this situation and yields a scaled and shifted version of the standard KDE. This scale and shift is especially evident in the preservation of the bump on the right hand side of Figure 4.

### 4.2 Datasets

In our remaining experiments we investigate two performance metrics for different amounts of contamination. We perform our experiments on 12 classification datasets (names given in the supplemental material) where the 0 label is used as the target density and the 1 label is the anomalous contamination. This experimental setup **does not** satisfy Assumption A. The training datasets are constructed with $n_0$ samples from label 0 and $\frac{\varepsilon}{1-\varepsilon}n_0$ samples from label 1, thus making an $\varepsilon$ proportion of our samples come from the contaminating density. For our experiments we use the values $\varepsilon = 0, 0.05, 0.1, 0.15, 0.20, 0.25, 0.30$. Given some dataset we are interested in how well our density estimators $\widehat{f}$ estimate the density of the 0 class of our dataset, $f_{tar}$. Each test is performed on 15 permutations of the dataset. The experimental setup here is similar to the setup in Kim & Scott [6], the most significant difference being that $\sigma$ is set differently.

### 4.3 Performance Criteria

First we investigate the Kullback-Leibler (KL) divergence

$$D_{KL}\left(\widehat{f}||f_0\right) = \int \widehat{f}(x) \log\left(\frac{\widehat{f}(x)}{f_0(x)}\right) dx.$$

This KL divergence is large when $\widehat{f}$ estimates $f_0$ to have mass where it does not. For example, in our context, $\widehat{f}$ makes mistakes because of outlying contamination. We estimate this KL divergence as follows. Since we do not have access to $f_0$, it is estimated from the testing sample using a KDE, $\widetilde{f_0}$. The bandwidth for $\widetilde{f_0}$ is set using the testing data with a LOOCV line search minimizing $D_{KL}\left(f_0||\widetilde{f_0}\right)$, which is described in more detail below. We then approximate the integral using a sample mean by generating samples from $\widehat{f}$, $\{x'_i\}_1^{n'}$ and using the estimate

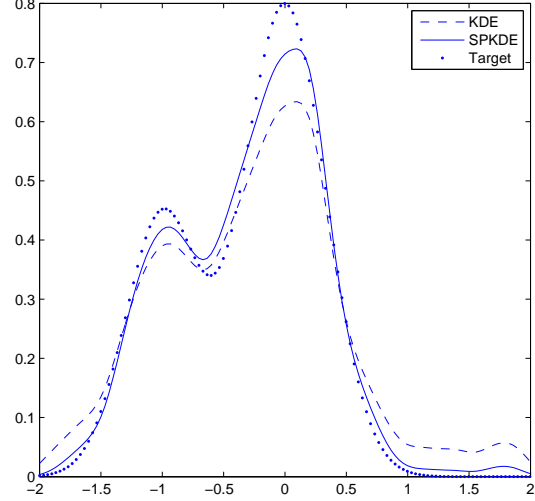

Figure 4: KDE and SPKDE in the presence of uniform noise

$$D_{KL}\left(\widehat{f}||f_0\right) \approx \frac{1}{n'} \sum_1^{n'} \log\left(\frac{\widehat{f}(x'_i)}{\widetilde{f_0}(x'_i)}\right).$$

The number of generated samples $n'$ is set to double the number of training samples.

Since KL divergence isn't symmetric we also investigate

$$D_{KL}\left(f_0||\widehat{f}\right) = \int f_0(x) \log\left(\frac{f_0(x)}{\widehat{f}(x)}\right) dx = C - \int f_0(y) \log\left(\widehat{f}(y)\right) dy,$$

where $C$ is a constant not depending on $\widehat{f}$. This KL divergence is large when $f_0$ has mass where $\widehat{f}$ does not. The final term is easy to estimate using expectation. Let $\{x''_i\}_1^{n''}$ be testing samples from $f_0$ (not used for training). The following is a reasonable approximation

$$-\int f_0(y) \log\left(\widehat{f}(y)\right) dy \approx -\frac{1}{n''} \sum_1^{n''} \log\left(\widehat{f}(x''_i)\right).$$

For a given performance metric and contamination amount, we compare the mean performance of two density estimators across datasets using the Wilcoxon signed rank test [17]. Given $N$ datasets we first rank the datasets according to the absolute difference between performance criterion, with $h_i$ being the rank of the $i$th dataset. For example if the $j$th dataset has the largest absolute difference we set $h_j = N$ and if the $k$th dataset has the smallest absolute difference we set $h_k = 1$. We let $R_1$ be the sum of the $h_i$s where method one's metric is greater than metric two's and $R_2$ be the sum of the $h_i$s where method two's metric is larger. The test statistic is $\min(R_1, R_2)$, which we do not report. Instead we report $R_1$ and $R_2$ and the $p$-value that the two methods do not perform the same on average. $R_i < R_j$ is indicative of method $i$ performing better than method $j$.

## 4.4 Methods

The data were preprocessed by scaling to fit in the unit cube. This scaling technique was chosen over whitening because of issues with singular covariance matrices. The Gaussian kernel was used for all density estimates. For each permutation of each dataset, the bandwidth parameter is set using the training data with a LOOCV line search minimizing $D_{KL}\left(f_{obs}||\widehat{f}\right)$, where $\widehat{f}$ is the KDE based on the contaminated data and $f_{obs}$ is the observed density. This metric was used in order to maximize the performance of the traditional KDE in KL divergence metrics. For the SPKDE the parameter $\beta$ was chosen to be 2 for all experiments. This choice of $\beta$ is based on a few preliminary experiments

Table 1: Wilcoxon signed rank test results

| | Wilcoxon Test Applied to $D_{KL}\left(\hat{f}||f_0\right)$ | | | | | | | Wilcoxon Test Applied to $D_{KL}\left(f_0||\hat{f}\right)$ | | | | | | |
|---|---|---|---|---|---|---|---|---|---|---|---|---|---|---|
| $\varepsilon$ | 0 | 0.05 | 0.1 | 0.15 | 0.2 | 0.25 | 0.3 | 0 | 0.05 | 0.1 | 0.15 | 0.2 | 0.25 | 0.3 |
| SPKDE | 5 | 0 | 1 | 2 | 0 | 0 | 0 | 37 | 30 | 27 | 21 | 17 | 16 | 17 |
| KDE | 73 | 78 | 77 | 76 | 78 | 78 | 78 | 41 | 48 | 51 | 57 | 61 | 62 | 61 |
| p-value | .0049 | 5e-4 | 1e-3 | .0015 | 5e-4 | 5e-4 | 5e-4 | .91 | .52 | .38 | .18 | .092 | .078 | .092 |
| SPKDE | 53 | 59 | 58 | 67 | 63 | 61 | 63 | 14 | 14 | 14 | 10 | 10 | 12 | 12 |
| RKDE | 25 | 19 | 20 | 11 | 15 | 17 | 15 | 64 | 64 | 64 | 68 | 68 | 66 | 66 |
| p-value | 0.31 | 0.13 | 0.15 | .027 | .064 | .092 | .064 | .052 | .052 | .052 | .021 | .021 | .034 | .034 |
| SPKDE | 0 | 0 | 1 | 1 | 0 | 2 | 0 | 29 | 21 | 19 | 15 | 13 | 9 | 11 |
| rejKDE | 78 | 78 | 77 | 77 | 78 | 76 | 78 | 49 | 57 | 59 | 63 | 65 | 69 | 67 |
| p-value | 5e-4 | 5e-4 | 1e-3 | 1e-3 | 5e-4 | .0015 | 5e-4 | .47 | .18 | .13 | .064 | .043 | .016 | .027 |

for which it yielded good results over various sample contamination amounts. The construction of the RKDE follows exactly the methods outlined in the "Experiments" section of Kim & Scott [6]. It is worth noting that the RKDE depends on the loss function used and that the Hampel loss used in these experiments very aggressively suppresses the kernel weights on the tails. Because of this we expect that RKDE performs well on the $D_{KL}\left(\hat{f}||f_0\right)$ metric. We also compare the SPKDE to a kernel density estimator constructed from samples declared non-anomalous by a level set anomaly detection as described in Section 2.3. To do this we first construct the classic KDE, $\bar{f}_\sigma^n$ and then reject those samples in the lower 10th percentile of $\bar{f}_\sigma^n(X_i)$. Those samples not rejected are used in a new KDE, the "rejKDE" using the same $\sigma$ parameter.

## 4.5 Results

We present the results of the Wilcoxon signed rank tests in Table 1. Experimental results for each dataset can be found in the supplemental material. From the results it is clear that the SPKDE is effective at compensating for contamination in the $D_{KL}\left(\hat{f}||f_0\right)$ metric, albeit not quite as well as the RKDE. The main advantage of the SPKDE over the RKDE is that it significantly outperforms the RKDE in the $D_{KL}\left(f_0||\hat{f}\right)$ metric. The rejKDE performs significantly worse than the SPKDE on almost every experiment. Remarkably the SPKDE outperforms the KDE in the situation with no contamination ($\varepsilon = 0$) for both performance metrics.

## 5 Conclusion

Robustness in the setting of nonparametric density estimation is a topic that has received little attention despite extensive study of robustness in the parametric setting. In this paper we introduced a robust version of the KDE, the SPKDE, and developed a new formalism for analysis of robust density estimation. With this new formalism we proposed a contamination model and decontaminating transform to recover a target density in the presence of noise. The contamination model allows that the target and contaminating densities have overlapping support and that the basic shape of the target density is not modified by the contaminating density. The proposed transform is computationally prohibitive to apply directly to the finite sample KDE and the SPKDE is used to approximate the transform. The SPKDE was shown to asymptotically converge to the desired transform. Experiments have shown that the SPKDE is more effective than the RKDE at minimizing $D_{KL}\left(f_0||\hat{f}\right)$.

Furthermore the p-values for these experiments were smaller than the p-values for the $D_{KL}\left(\hat{f}||f_0\right)$ experiments where the RKDE outperforms the SPKDE.

### Acknowledgements

This work support in part by NSF Awards 0953135, 1047871, 1217880, 1422157. We would also like to thank Samuel Brodkey for his assistance with the simulation code.

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
