[Supplementary Material · sup.pdf]

# Supplemental Material for Robust Kernel Density Estimation by Scaling and Projection in Hilbert Space

**Robert A. Vandermeulen**
Department of EECS
University of Michigan
Ann Arbor, MI 48109
rvdm@umich.edu

**Clayton D. Scott**
Deparment of EECS
Univeristy of Michigan
Ann Arbor, MI 48109
clayscot@umich.edu

## 1 Proofs

*Proof of Lemma 1 and 2.* We will prove Lemma 1 and 2 simultaneously. The $f$ in Lemma 1 and Lemma 2 are the same and all notation is consistent between the two lemmas. First we will show that $\|g_{\alpha,\beta}\|_{L^1}$ is continuous in $\alpha$. Let $\{a_i\}_1^\infty$ be a non-negative sequence in $\mathbb{R}$ converging to arbitrary $a \geq 0$. Since $g_{a_i,\beta}$ is dominated by $\beta f$ and $g_{a_i,\beta}$ converges to $g_{a,\beta}$ pointwise, by the dominated convergence theorem we know $\|g_{a_i,\beta}\|_{L^1} \to \|g_{a,\beta}\|_{L^1}$, thus proving the continuity of $\|g_{\alpha,\beta}\|_{L^1}$. Since $\|g_{0,\beta}\|_{L^1} = \beta > 1$ and $\|g_{\alpha,\beta}\|_{L^1} \to 0$ as $\alpha \to \infty$, by the intermediate value theorem there exists $\alpha'$ such that $\|g_{\alpha',\beta}\|_{L^1} = 1$. This proves the existence part of Lemma 1. Let $\widetilde{f}_\beta = g_{\alpha',\beta}$. Clearly $\mathcal{D}$ is convex so the closure (in $L^2$) $\bar{\mathcal{D}}$ is also convex. Since $\bar{\mathcal{D}}$ is a closed and convex set in a Hilbert space, $\arg\min_{g \in \bar{\mathcal{D}}} \|g - \beta f\|_{L^2}$ admits a unique minimizer. Note that $\widetilde{f}_\beta$ being the unique minimizer is equivalent to showing that, for all $c$ in $\bar{\mathcal{D}}$ (Theorem 3.14 in [1])

$$\left\langle c - \widetilde{f}_\beta, \beta f - \widetilde{f}_\beta \right\rangle \leq 0.$$

Because this is continuous over the $c$ term and $\mathcal{D}$ is dense in $\bar{\mathcal{D}}$ we need only show that the inequality holds over all $c \in \mathcal{D}$. To this end, note that for all $x$,

$$\beta f(x) - \max\{0, \beta f(x) - \alpha'\} \leq \alpha'$$

and that if $\widetilde{f}_\beta(x) > 0$ then

$$\widetilde{f}_\beta(x) = \beta f(x) - \alpha'.$$

From this we have

$$
\begin{aligned}
&\left\langle c - \widetilde{f}_\beta, \beta f - \widetilde{f}_\beta \right\rangle \\
&= \left\langle c, \beta f - \widetilde{f}_\beta \right\rangle - \left\langle \widetilde{f}_\beta, \beta f - \widetilde{f}_\beta \right\rangle \\
&= \int c(x) \left( \beta f(x) - \widetilde{f}_\beta(x) \right) dx \\
&\quad - \int \widetilde{f}_\beta(x) \left( \beta f(x) - \widetilde{f}_\beta(x) \right) dx \\
&\leq \int c(x) \alpha' dx \\
&\quad - \int \widetilde{f}_\beta(x) \left( \beta f(x) - (\beta f(x) - \alpha') \right) dx \\
&= \alpha' - \alpha' \\
&= 0.
\end{aligned}
$$

From this we get that $\widetilde{f}_\beta$ is the unique minimizer. If there existed $\alpha'' \neq \alpha'$ such that $g_{\alpha'',\beta}$ was also a pdf, then there would be two minimizers of $\arg\min_{g\in\bar{\mathcal{D}}}\|g - \beta f\|_{L^2}$, which is impossible since the minimizer is unique, thus proving the uniqueness of $\alpha'$. □

*Proof of Proposition 3.* In this proof we will be working with a hypothetical $f_{tar}$ and $f_{con}$ in $\mathcal{D}$. Define "Assumption B" to be that there exists two sets $S \subset \text{supp}(f_{tar})$ and $T \subset \mathbb{R}^d$, which have nonzero Lebesgue measure, such that $f_{con}(T) > f_{con}(S)$. We will now show that Assumption A not holding is equivalent to Assumption B.

A⇒not B: Let $S \subset \text{supp}(f_{tar})$ and $T \subset \mathbb{R}^d$ both have nonzero Lebesgue measure. From Assumption A we know for Lebesgue almost all $s \in S$ that $f_{con}(s) = u$, for some $u$ and $f_{con}(T) \leq u$ Lebesgue almost everywhere.

not A⇒B: If Assumption A is not satisfied either $f_{con}$ is not almost Lebesgue everywhere uniform over $\text{supp}(f_{tar})$ or $f_{con}$ is Lebesgue almost everywhere uniform on $\text{supp}(f_{tar})$ with value $u$ but there exists some set $Q \subset \mathbb{R}^d$ of nonzero Lebesgue measure such that $f_{con}(Q) > u$. Both of these situations clearly imply Assumption B.

This proves that the negation of Assumption A is Assumption B.

Let $f_{con}$ and $f_{tar}$ satisfy Assumption B and $\varepsilon \in (0,1)$ be arbitrary. By Lemma 1 we know there exists a unique $\alpha$ such that $\max\left\{\frac{1}{1-\varepsilon}\left((1-\varepsilon)f_{tar}(\cdot) + \varepsilon f_{con}\right) - \alpha, 0\right\}$ is a pdf. First we will show that $\alpha < \text{ess sup}_x \frac{\varepsilon}{1-\varepsilon}f_{con}(x)$. If $\text{ess sup}_x \frac{\varepsilon}{1-\varepsilon}f_{con}(x) = \infty$ then clearly $\alpha < \text{ess sup}_x \frac{\varepsilon}{1-\varepsilon}f_{con}(x)$. Let $r = \text{ess sup}_x \frac{\varepsilon}{1-\varepsilon}f_{con}(x) < \infty$. Let $S, T \subset \mathbb{R}^d$ satisfy the properties in the definition of Assumption B. Observe that

$$\int \max\left\{\frac{1}{1-\varepsilon}\left((1-\varepsilon)f_{tar}(x) + \varepsilon f_{con}(x)\right) - r, 0\right\}dx$$
$$= \int \max\left\{f_{tar}(x) + \frac{\varepsilon}{1-\varepsilon}f_{con}(x) - r, 0\right\}dx$$
$$= \int_S \max\left\{f_{tar}(x) + \frac{\varepsilon}{1-\varepsilon}f_{con}(x) - r, 0\right\}dx + \int_{S^C}\max\left\{f_{tar}(x) + \frac{\varepsilon}{1-\varepsilon}f_{con}(x) - r, 0\right\}dx.$$

Note that on the set $S$ we have that $\max\left\{f_{tar}(x) + \frac{\varepsilon}{1-\varepsilon}f_{con}(x) - r, 0\right\} < f_{tar}$. Now we have

$$\int_S \max\left\{f_{tar}(x) + \frac{\varepsilon}{1-\varepsilon}f_{con}(x) - r, 0\right\}dx + \int_{S^C}\max\left\{f_{tar}(x) + \frac{\varepsilon}{1-\varepsilon}f_{con}(x) - r, 0\right\}dx$$
$$< \int_S f_{tar}(x)dx + \int_{S^C}f_{tar}(x)dx$$
$$< 1$$

and thus $\alpha < r$ (i.e. the cutoff value for $R_\varepsilon^A(f_{obs})$ is lower than the essential supremum of $f_{con}$). Because $\alpha < \text{ess sup}_x \frac{\varepsilon}{1-\varepsilon}f_{con}(x)$, on the set for which $\frac{\varepsilon}{1-\varepsilon}f_{con}(\cdot) > \alpha$ (which has nonzero Lebesgue measure) we have that $\max\left\{f_{tar}(\cdot) + \frac{\varepsilon}{1-\varepsilon}f_{con} - \alpha, 0\right\} > f_{tar}$, so $\max\left\{f_{tar}(\cdot) + \frac{\varepsilon}{1-\varepsilon}f_{con} - \alpha, 0\right\} \neq f_{tar}$. □

*Proof of Theorem 1.* Given a set $S \subset L^2\left(\mathbb{R}^d\right)$ let $P_S$ be the projection operator onto $S$. Consider the following decomposition

$$\left\|f_{\sigma,\beta}^n - f_\beta'\right\|_{L^2} = \left\|P_{\mathcal{D}_\sigma^n}\beta\bar{f}_\sigma^n - P_{\bar{\mathcal{D}}}\beta f\right\|_{L^2}$$
$$\leq \left\|P_{\mathcal{D}_\sigma^n}\beta\bar{f}_\sigma^n - P_{\mathcal{D}_\sigma^n}\beta f\right\|_{L^2} + \left\|P_{\mathcal{D}_\sigma^n}\beta f - P_{\bar{\mathcal{D}}}\beta f\right\|_{L^2}$$

Note that we are projecting onto $\bar{\mathcal{D}}$ rather than $\mathcal{D}$ does not matter as was shown in the proof of Lemma 1 and 2. Furthermore note that $f_\beta' = P_{\bar{\mathcal{D}}}\beta f$. The projection operator onto a closed convex set is Lipschitz continuous with constant 1 (Proposition 4.8 in [1]) so the first term goes to zero by

standard KDE consistency (which we prove later). Convergence of the second term is a bit more involved. First we will show that $\left\|P_{\mathcal{D}_\sigma^n}\beta f - \beta f\right\|_{L^2} \xrightarrow{p} \left\|P_{\bar{\mathcal{D}}}\beta f - \beta f\right\|_{L^2}$, and then we will show that this implies $\left\|P_{\mathcal{D}_\sigma^n}\beta f - f_\beta'\right\|_{L^2} \xrightarrow{p} 0$.

We know $\mathcal{D}_\sigma^n \subset \bar{\mathcal{D}}$ so $\left\|P_{\mathcal{D}_\sigma^n}\beta f - \beta f\right\|_{L^2} \geq \left\|P_{\bar{\mathcal{D}}}\beta f - \beta f\right\|_{L^2}$. We also know that for all $\delta \in \mathcal{D}_\sigma^n$, $\left\|P_{\mathcal{D}_\sigma^n}\beta f - \beta f\right\|_{L^2} \leq \|\delta - \beta f\|_{L^2}$. Because of these two facts, in order to show $\left\|P_{\mathcal{D}_\sigma^n}\beta f - \beta f\right\|_{L^2} \xrightarrow{p} \left\|P_{\bar{\mathcal{D}}}\beta f - \beta f\right\|_{L^2}$, it is sufficient to find a sequence $\{g_\sigma^n\} \subset \mathcal{D}_\sigma^n$ such that $\left\|g_\sigma^n - f_\beta'\right\|_{L^2} \xrightarrow{p} 0$. Since $\beta f > f_\beta'$ we can generate $g_\sigma^n$ by applying rejection sampling to $X_1, \ldots, X_n$ to generate a subsample $X_1', \ldots, X_{m_n}'$ which are iid from $f_\beta'$. For all $i$ the event of $X_i$ getting rejected is independent with equal probability. The probability of a sample not being rejected is greater than zero so there exists a $b > 0$ such that $\mathbb{E}[m_n] > bn$. From this and the strong law of large numbers we have that $\mathbb{P}\left(m_n\sigma^d \to \infty\right) = 1$. Using this subsample we can construct $g_\sigma^n \triangleq \frac{1}{m_n}\sum_1^{m_n} k_\sigma\left(\cdot, X_i'\right) \in \mathcal{D}_\sigma^n$ which is a KDE of $f_\beta'$, so by standard KDE consistency $\left\|f_\beta' - g_\sigma^n\right\|_{L^2} \xrightarrow{p} 0$, and thus $\left\|P_{\mathcal{D}_\sigma^n}\beta f - \beta f\right\|_{L^2} \xrightarrow{p} \left\|P_{\bar{\mathcal{D}}}\beta f - \beta f\right\|_{L^2}$.

Let $\widetilde{f}_{\sigma,\beta}^n \triangleq P_{\mathcal{D}_\sigma^n}\beta f$. Finally we are going to show that $\left\|P_{\mathcal{D}_\sigma^n}\beta f - \beta f\right\|_{L^2} \xrightarrow{p} \left\|P_{\bar{\mathcal{D}}}\beta f - \beta f\right\|_{L^2}$ implies that $\left\|\widetilde{f}_{\sigma,\beta}^n - f_\beta'\right\|_{L^2} \xrightarrow{p} 0$. The functional $\|\beta f - \cdot\|_{L^2}^2$ is strongly convex with convexity constant 2 (Example 10.7 in [1]). This means that for any $a \in (0,1)$, we have

$$\left\|\beta f - \left(a\widetilde{f}_{\sigma,\beta}^n + (1-a)f_\beta'\right)\right\|_{L^2}^2 + a(1-a)\left\|\widetilde{f}_{\sigma,\beta}^n - f_\beta'\right\|_{L^2}^2$$
$$\leq a\left\|\beta f - \widetilde{f}_{\sigma,\beta}^n\right\|_{L^2}^2 + (1-a)\left\|\beta f - f_\beta'\right\|_{L^2}^2.$$

Letting $a = 1/2$ gives us

$$\left\|\beta f - \frac{\widetilde{f}_{\sigma,\beta}^n + f_\beta'}{2}\right\|_{L^2}^2 + \frac{1}{4}\left\|\widetilde{f}_{\sigma,\beta}^n - f_\beta'\right\|_{L^2}^2 \leq \frac{1}{2}\left\|\beta f - \widetilde{f}_{\sigma,\beta}^n\right\|_{L^2}^2 + \frac{1}{2}\left\|\beta f - f_\beta'\right\|_{L^2}^2$$

Since

$$\left\|\beta f - f_\beta'\right\|_{L^2}^2 \leq \left\|\beta f - \widetilde{f}_{\sigma,\beta}^n\right\|_{L^2}^2$$

and

$$\left\|\beta f - f_\beta'\right\|_{L^2}^2 \leq \left\|\beta f - \frac{\widetilde{f}_{\sigma,\beta}^n + f_\beta'}{2}\right\|_{L^2}^2$$

we have

$$\left\|\beta f - f_\beta'\right\|_{L^2}^2 + \frac{1}{4}\left\|\widetilde{f}_{\sigma,\beta}^n - f_\beta'\right\|_{L^2}^2 \leq \left\|\beta f - \widetilde{f}_{\sigma,\beta}^n\right\|_{L^2}^2$$

or equivalently

$$\left\|\widetilde{f}_{\sigma,\beta}^n - f_\beta'\right\|_{L^2}^2 \leq 4\left(\left\|\beta f - \widetilde{f}_{\sigma,\beta}^n\right\|_{L^2}^2 - \left\|\beta f - f_\beta'\right\|_{L^2}^2\right).$$

The right side of the last equation goes to zero in probability, thus finishing our proof. $\qquad\square$

*Proof of KDE $L^2$ consistency.* Let $\bar{f}_\sigma = \mathbb{E}[k_\sigma\left(\cdot, X_i\right)] = \int k_\sigma\left(\cdot, x\right)g(x)\,dx$. Using the triangle inequality we have

$$\left\|f - \bar{f}_\sigma^n\right\|_{L^2} \leq \left\|f - \bar{f}_\sigma\right\|_{L^2} + \left\|\bar{f}_\sigma - \bar{f}_\sigma^n\right\|_{L^2}.$$

The left summand goes to zero as $\sigma \to 0$ by elementary analysis (see Theorem 8.14 in [2]). To take care of the right side with use the following lemma which is a Hilbert space version of Hoeffding's inequality from Steinwart & Christmann [3], Corollary 6.15.

**Lemma** (**Hoeffding's inequality in Hilbert space**). *Let $(\Omega, \mathcal{A}, P)$ be a probability space, $H$ be a separable Hilbert space, and $B > 0$. Furthermore, let $\xi_1, \dots \xi_n : \Omega \to H$ be independent $H$-valued random variables satisfying $\|\xi_i\|_\infty \leq B$ for all $i$. Then, for all $\tau > 0$, we have*

$$P\left( \left\| \frac{1}{n} \sum_1^n (\xi_i - \mathbb{E}\left[\xi_i\right]) \right\|_H \geq B\sqrt{\frac{2\tau}{n}} + B\sqrt{\frac{1}{n}} + \frac{4B\tau}{3n} \right) \leq e^{-\tau}..$$

Note that $\|\xi_i\|_\infty = \operatorname{ess\,sup}_{\omega \in \Omega} \|\xi_i(\omega)\|_H$. Plugging in $\xi_i = k_\sigma(\cdot, X_i)$ we get

$$P\bigg( \left\| \bar{f}_\sigma^n - \bar{f}_\sigma \right\|_{L^2} \geq \|k_\sigma(\cdot, X_i)\|_{L^2} \sqrt{\frac{2\tau}{n}}$$
$$+ \|k_\sigma(\cdot, X_i)\|_{L^2} \sqrt{\frac{1}{n}} + \frac{4\|k_\sigma(\cdot, X_i)\|_{L^2} \tau}{3n} \bigg) \leq e^{-\tau}.$$

It is straightforward to show that there exists $Q > 0$ such that $\|k_\sigma(\cdot, X_i)\|_{L^2} = Q\sigma^{-d/2}$, giving us

$$P\bigg( \left\| \bar{f}_\sigma^n - \bar{f}_\sigma \right\|_{L^2} \geq Q\sigma^{-d/2}\sqrt{\frac{2\tau}{n}}$$
$$+ Q\sigma^{-d/2}\sqrt{\frac{1}{n}} + \frac{4Q\sigma^{-d/2}\tau}{3n} \bigg) \leq e^{-\tau}.$$

Letting $n\sigma^d \to \infty$ sends all of the summands in the previous expression to zero for fixed $\tau$. Because of this there exists a positive sequence $\{\tau_i\}_1^\infty$ such that $\tau_i \to \infty$ and but increases slowly enough that $Q\sigma^{-d/2}\sqrt{\frac{2\tau_n}{n}} + Q\sigma^{-d/2}\sqrt{\frac{1}{n}} + \frac{4Q\sigma^{-d/2}\tau_n}{3n} \to 0$ as $n \to \infty$, where $\sigma$ depends implicitly on $n$. From this it is clear that $\left\| \bar{f}_\sigma^n - \bar{f}_\sigma \right\|_{L^2} \xrightarrow{p} 0$. $\qquad \square$

*Proof of Corollary 1.* Let $\lambda$ be the Lebesgue measure. Let $S \subset \mathbb{R}^d$ be such that $\lambda(S) < \infty$. By Hölders inequality we have

$$\begin{aligned} \left\| (f'_\beta - f^n_{\sigma,\beta}) \chi_S \right\|_{L^1} &< \left\| f'_\beta - f^n_{\sigma,\beta} \right\|_{L^2} \|\chi_S\|_{L^2} \\ &= \left\| f'_\beta - f^n_{\sigma,\beta} \right\|_{L^2} \sqrt{\lambda(S)}. \end{aligned}$$

From this we have that, that $f^n_{\sigma,\beta}$ converges in probability to $f'_\beta$ in $L^1$ norm, when restricted to a set of finite Lebesgue measure. Let $\delta > 0$ be arbitrary. Choose $S$ to be a set of finite measure large enough that $\int_{S^C} f'_\beta(x)\,dx < \delta/8$. Note that this implies $\left\| f'_\beta \chi_S \right\|_{L^1} \geq \frac{7}{8}\delta$, a fact we will use later. Notice that

$$\left\| f'_\beta - f^n_{\sigma,\beta} \right\|_{L^1} = \left\| (f'_\beta - f^n_{\sigma,\beta}) \chi_S \right\|_{L^1} + \left\| (f'_\beta - f^n_{\sigma,\beta}) \chi_{S^C} \right\|_{L^1}.$$

We have already shown that the left summand in the converges in probability to zero, so it becomes bounded by $\delta/8$ with probability going to one. To finish the proof we need only show that the right summand is bounded by $\frac{7}{8}\delta$ with probability going to one. Using the triangle inequality we have

$$\begin{aligned} \left\| (f'_\beta - f^n_{\sigma,\beta}) \chi_{S^C} \right\|_{L^1} &\leq \left\| f'_\beta \chi_{S^C} \right\|_1 + \left\| f^n_{\sigma,\beta} \chi_{S^C} \right\|_{L^1} \\ &< \delta/8 + \left\| f^n_{\sigma,\beta} \chi_{S^C} \right\|_{L^1}. \end{aligned}$$

Now it is sufficient to show that $\left\| f^n_{\sigma,\beta} \chi_{S^C} \right\|_1$ becomes bounded by $\frac{3}{4}\delta$ with probability going to one. To finish the proof,

$$\left\| f^n_{\sigma,\beta} \chi_S \right\|_{L^1} + \left\| f^n_{\sigma,\beta} \chi_{S^C} \right\|_{L^1} = 1$$

therefore

$$\left\| f^n_{\sigma,\beta} \chi_{S^C} \right\|_{L^1} = 1 - \left\| f^n_{\sigma,\beta} \chi_S \right\|_{L^1}$$

and we know that $\left\| f^n_{\sigma,\beta} \chi_S \right\|_{L^1} \xrightarrow{p} \left\| f'_\beta \chi_S \right\|_{L^1} \geq \frac{7}{8}\delta$ so with probability going to one $\left\| f^n_{\sigma,\beta} \chi_S \right\|_{L^1} \geq \delta/2$ and thus $\left\| f^n_{\sigma,\beta} \chi_{S^C} \right\|_{L^1} < \delta/2$. $\qquad \square$

*Proof of Theorem 2.* By the triangle inequality we have $\left\| f_{\sigma,\beta}^n - f_{tar} \right\|_{L^1} \leq \left\| f_{\sigma,\beta}^n - f'_\beta \right\|_{L^1} + \left\| f'_\beta - f_{tar} \right\|_{L^1}$. The left summand in the previous inequality goes to zero by Corollary 1, so it is sufficient to show that the right term is zero. The rest of this proof will effectively prove Proposition 2. Again let $g_{\alpha,\beta}(\cdot) = \max\{0, \beta f_{obs}(\cdot) - \alpha\}$. From Assumption A we know that Lebesgue almost everywhere on the support of $f_{tar}$, that $f_{con}$ is equal to some value $u$ and that $f_{con}$ is less than or equal to $u$ Lebesgue almost everywhere on $\mathbb{R}^d$. We will show that, $\alpha' = \frac{\varepsilon u}{1-\varepsilon}$, gives us $g_{\alpha',\beta} = f_{tar}$ which, by Lemma 1, implies $f_{tar} = f'_\beta$. Let $K$ be the support of $f_{tar}$.

First consider $x \in K^C$. Almost everywhere on $K^C$ have

$$
\begin{aligned}
g_{\alpha',\beta}(x) &= \max\left\{0, \beta f_{obs}(x) - \frac{\varepsilon u}{1-\varepsilon}\right\} \\
&= \max\left\{0, \frac{1}{1-\varepsilon} f_{con}(x)\varepsilon - \frac{\varepsilon u}{1-\varepsilon}\right\} \\
&\leq \max\left\{0, \frac{1}{1-\varepsilon} u\varepsilon - \frac{\varepsilon u}{1-\varepsilon}\right\} \\
&= 0.
\end{aligned}
$$

So $g_{\alpha',\beta}$ is zero almost everywhere not on the support of $f_{tar}$. Now let $x \in K$, then Lebesgue almost everywhere in $K$ we have

$$
\begin{aligned}
&g_{\alpha',\beta}(x) \\
&= \max\left\{0, \beta f_{obs}(x) - \frac{\varepsilon u}{1-\varepsilon}\right\} \\
&= \max\left\{0, \frac{1}{1-\varepsilon}\left((1-\varepsilon) f_{tar}(x) + f_{con}(x)\varepsilon\right) - \frac{\varepsilon u}{1-\varepsilon}\right\} \\
&= \max\left\{0, \frac{1}{1-\varepsilon}\left((1-\varepsilon) f_{tar}(x) + u\varepsilon\right) - \frac{\varepsilon u}{1-\varepsilon}\right\} \\
&= \max\left\{0, f_{tar}(x) + \frac{\varepsilon u}{1-\varepsilon} - \frac{\varepsilon u}{1-\varepsilon}\right\} \\
&= f_{tar}(x).
\end{aligned}
$$

From this we have that $g_{\alpha',\beta} = f_{tar}$ which is a pdf, which by Lemma 1 is therefore equal to $f'_\beta$. $\square$

# 2 Experimental Results

Table 1: Mean and Standard Deviation of $D_{KL}\left(\widehat{f}\|f_0\right)$

| Dataset | Algorithm | $\varepsilon$ | | | | | | |
|---|---|---|---|---|---|---|---|---|
| | | 0.00 | 0.05 | 0.10 | 0.15 | 0.20 | 0.25 | 0.30 |
| banana | SPKDE | 0.19±0.04 | 0.15±0.03 | 0.14±0.03 | 0.17±0.07 | 0.23±0.08 | 0.35±0.1 | 0.51±0.2 |
| | KDE | 0.19±0.1 | 0.32±0.1 | 0.53±0.2 | 0.66±0.2 | 0.84±0.2 | 1.1±0.2 | 1.2±0.2 |
| | RKDE | 0.81±0.3 | 0.78±0.3 | 0.77±0.3 | 0.71±0.4 | 0.61±0.3 | 0.63±0.3 | 0.66±0.3 |
| | rejKDE | 0.19±0.2 | 0.35±0.2 | 0.52±0.2 | 0.7±0.2 | 0.84±0.2 | 1.1±0.2 | 1.3±0.2 |
| breast-cancer | SPKDE | 3.2±0.7 | 3.4±0.8 | 3.2±0.8 | 3.5±0.9 | 3.7±1 | 3.9±1 | 4.2±1 |
| | KDE | 4±0.9 | 4.1±1 | 4±1 | 4.3±1 | 4.6±1 | 4.8±1 | 5±1 |
| | RKDE | 3.1±0.7 | 3.2±0.7 | 3±0.5 | 3.2±0.6 | 3.5±0.8 | 3.7±0.9 | 4±0.9 |
| | rejKDE | 4±0.8 | 4.1±1 | 4.1±1 | 4.3±1 | 4.6±1 | 4.8±1 | 4.9±1 |
| diabetis | SPKDE | 0.8±0.05 | 0.84±0.09 | 0.8±0.1 | 0.84±0.1 | 0.87±0.1 | 0.91±0.08 | 0.89±0.09 |
| | KDE | 1.5±0.2 | 1.6±0.3 | 1.8±0.3 | 1.8±0.4 | 1.9±0.4 | 2±0.3 | 2±0.4 |
| | RKDE | 0.99±0.1 | 1±0.1 | 0.96±0.1 | 0.98±0.1 | 1±0.1 | 1±0.1 | 0.98±0.1 |
| | rejKDE | 1.5±0.2 | 1.6±0.2 | 1.8±0.4 | 1.9±0.5 | 1.9±0.5 | 2±0.4 | 2.1±0.5 |
| german | SPKDE | 6.6±0.9 | 6.8±1 | 6.9±0.9 | 7±0.9 | 6.9±1 | 7.2±0.7 | 7.4±0.7 |
| | KDE | 7±1 | 7±1 | 7.3±0.9 | 7.4±1 | 7.4±1 | 7.6±0.8 | 7.8±0.8 |
| | RKDE | 5.4±0.7 | 5.6±0.8 | 5.8±0.7 | 5.8±0.8 | 5.9±0.8 | 6±0.7 | 6.2±0.6 |
| | rejKDE | 7±1 | 7.2±1 | 7.4±1 | 7.5±1 | 7.5±1 | 7.7±0.8 | 7.8±0.7 |
| heart | SPKDE | 4±0.7 | 4±0.9 | 4.2±0.7 | 4.5±0.8 | 4.8±1 | 5.1±1 | 5.1±1 |
| | KDE | 4.7±1 | 5.1±1 | 5.3±1 | 5.6±1 | 5.8±1 | 6.2±1 | 6.6±1 |
| | RKDE | 3.8±0.9 | 3.8±0.8 | 3.9±0.6 | 4.2±0.8 | 4.2±0.9 | 4.5±1 | 4.9±1 |
| | rejKDE | 4.8±0.9 | 5.3±1 | 5.2±1 | 5.6±1 | 5.6±1 | 6.3±1 | 6.4±1 |
| ionosphere scale | SPKDE | 13±2 | 13±2 | 13±2 | 13±2 | 12±2 | 11±2 | 11±1 |
| | KDE | 15±2 | 14±2 | 14±2 | 15±2 | 14±2 | 13±2 | 14±2 |
| | RKDE | 10±2 | 10±2 | 9.9±2 | 9.2±2 | 8±3 | 6.7±2 | 7.5±3 |
| | rejKDE | 16±2 | 15±2 | 15±2 | 14±1 | 14±2 | 14±2 | 14±2 |
| ringnorm | SPKDE | 4.8±0.4 | 5.3±0.9 | 6.3±1 | 7.3±1 | 8±1 | 9.2±1 | 9±0.9 |
| | KDE | 4.9±0.4 | 5.7±0.9 | 7.4±1 | 8.6±1 | 11±2 | 13±2 | 14±0.7 |
| | RKDE | 4.4±0.2 | 3.8±0.6 | 4±0.6 | 4.1±0.6 | 4.7±1 | 5.7±0.6 | 6.1±0.5 |
| | rejKDE | 5±0.3 | 5.8±0.8 | 7.3±1 | 8.5±1 | 10±2 | 13±1 | 14±0.8 |
| sonar scale | SPKDE | 30±7 | 31±8 | 30±8 | 33±7 | 33±7 | 33±7 | 35±7 |
| | KDE | 31±6 | 31±9 | 31±8 | 32±8 | 34±7 | 35±8 | 35±8 |
| | RKDE | 32±9 | 32±7 | 32±7 | 31±7 | 33±8 | 34±7 | 35±7 |
| | rejKDE | 31±9 | 32±8 | 32±9 | 34±7 | 33±8 | 33±7 | 36±8 |
| splice | SPKDE | 21±0.3 | 21±0.2 | 21±0.3 | 21±0.3 | 21±0.2 | 21±0.2 | 20±0.4 |
| | KDE | 21±0.3 | 21±0.2 | 21±0.2 | 21±0.3 | 21±0.3 | 21±0.2 | 20±0.2 |
| | RKDE | 21±0.5 | 21±0.5 | 21±0.6 | 21±0.4 | 21±0.4 | 20±0.6 | 20±0.6 |
| | rejKDE | 21±0.3 | 21±0.3 | 21±0.2 | 21±0.2 | 21±0.3 | 21±0.2 | 20±0.2 |
| thyroid | SPKDE | 0.59±0.2 | 0.69±0.4 | 1.1±0.8 | 1.3±0.8 | 1.2±0.7 | 1.1±0.7 | 1.3±0.6 |
| | KDE | 0.6±0.2 | 4.5±3 | 11±7 | 16±7 | 20±7 | 22±5 | 32±8 |
| | RKDE | 0.56±0.1 | 0.88±0.5 | 1.3±0.9 | 1.6±1 | 1.5±0.8 | 1.3±0.6 | 1.4±0.8 |
| | rejKDE | 0.59±0.2 | 4.9±3 | 8.6±5 | 17±6 | 22±9 | 25±7 | 33±8 |
| twonorm | SPKDE | 4.8±0.4 | 4.6±0.5 | 4.6±0.5 | 4.8±0.7 | 5±0.9 | 5.4±0.9 | 6.2±1 |
| | KDE | 4.8±0.4 | 4.8±0.5 | 4.9±0.5 | 5.1±0.6 | 5.2±0.9 | 5.7±0.9 | 6.6±1 |
| | RKDE | 4.2±0.4 | 3.8±0.4 | 3.9±0.5 | 4±0.5 | 4.1±0.7 | 4.7±0.9 | 5.5±0.8 |
| | rejKDE | 4.9±0.5 | 4.7±0.6 | 4.9±0.5 | 5±0.7 | 5.2±0.8 | 5.7±0.9 | 6.6±1 |
| waveform | SPKDE | 4.8±0.8 | 4.8±0.8 | 5.2±1 | 5.6±0.9 | 6.1±0.8 | 6.2±0.8 | 6.7±0.5 |
| | KDE | 5±0.7 | 4.9±0.7 | 5.3±1 | 5.7±1 | 6.3±0.9 | 6.2±0.8 | 6.8±0.4 |
| | RKDE | 4.5±0.7 | 4.4±0.6 | 4.7±0.9 | 5.2±1 | 5.6±0.8 | 5.7±0.7 | 6.1±0.4 |
| | rejKDE | 4.9±0.7 | 4.9±0.7 | 5.4±1 | 5.8±0.9 | 6.2±0.9 | 6.3±0.8 | 6.8±0.4 |

Table 2: Mean and Standard Deviation of $D_{KL}\left(f_0||\widehat{f}\right)$

| Dataset | Algorithm | $\varepsilon$ | | | | | | |
|---|---|---|---|---|---|---|---|---|
| | | 0.00 | 0.05 | 0.10 | 0.15 | 0.20 | 0.25 | 0.30 |
| banana | SPKDE | -0.57±0.2 | -0.69±0.2 | -0.73±0.2 | -0.78±0.2 | -0.81±0.2 | -0.79±0.2 | -0.75±0.2 |
| | KDE | -0.85±0.2 | -0.83±0.2 | -0.8±0.1 | -0.8±0.1 | -0.8±0.1 | -0.77±0.1 | -0.74±0.1 |
| | RKDE | 15±1e+01 | 12±9 | 11±9 | 8.6±9 | 5.7±7 | 6.5±9 | 7.1±9 |
| | rejKDE | -0.73±0.2 | -0.8±0.2 | -0.8±0.2 | -0.82±0.1 | -0.82±0.1 | -0.79±0.1 | -0.75±0.1 |
| breast-cancer | SPKDE | -1.7±0.7 | -1.8±0.7 | -2±0.6 | -2±0.6 | -2.2±0.6 | -2.4±0.6 | -2.6±0.7 |
| | KDE | -1.8±0.7 | -1.9±0.6 | -2.1±0.6 | -2.1±0.6 | -2.3±0.6 | -2.4±0.6 | -2.6±0.7 |
| | RKDE | 2.2±2 | 1.8±3 | 1.4±2 | 0.77±2 | 0.29±2 | -0.025±2 | -0.43±2 |
| | rejKDE | 0.4±2 | 0.1±2 | -0.35±2 | -0.69±1 | -1±1 | -1.2±1 | -1.4±1 |
| diabetis | SPKDE | -3.4±0.8 | -3.7±0.7 | -4±0.6 | -4.2±0.6 | -4.5±0.5 | -4.6±0.4 | -4.8±0.5 |
| | KDE | -3.9±0.5 | -4.1±0.5 | -4.3±0.4 | -4.4±0.3 | -4.6±0.4 | -4.7±0.3 | -5±0.3 |
| | RKDE | -1.3±1 | -1.7±2 | -1.7±1 | -2±1 | -2.1±2 | -2.6±2 | -2.5±1 |
| | rejKDE | -3.7±0.7 | -3.9±0.6 | -4.2±0.5 | -4.3±0.4 | -4.5±0.4 | -4.6±0.4 | -4.9±0.4 |
| german | SPKDE | -0.067±0.4 | -0.15±0.4 | -0.21±0.4 | -0.26±0.4 | -0.32±0.4 | -0.41±0.4 | -0.48±0.4 |
| | KDE | -0.043±0.4 | -0.12±0.4 | -0.19±0.4 | -0.23±0.4 | -0.29±0.4 | -0.38±0.4 | -0.45±0.4 |
| | RKDE | 0.71±0.5 | 0.62±0.5 | 0.56±0.7 | 0.52±0.6 | 0.45±0.6 | 0.35±0.6 | 0.29±0.6 |
| | rejKDE | 0.26±0.5 | 0.16±0.5 | 0.07±0.5 | 0.039±0.5 | -0.026±0.5 | -0.12±0.5 | -0.2±0.5 |
| heart | SPKDE | 0.7±0.7 | 0.44±0.9 | 0.17±0.7 | 0.071±0.7 | -0.044±0.8 | -0.21±0.8 | -0.32±0.8 |
| | KDE | 0.71±0.7 | 0.46±0.8 | 0.2±0.7 | 0.12±0.7 | 0.0049±0.8 | -0.15±0.8 | -0.26±0.7 |
| | RKDE | 2.4±1 | 1.9±0.9 | 1.5±0.8 | 1.4±1 | 1.2±0.8 | 1±0.9 | 0.82±0.8 |
| | rejKDE | 1.3±0.9 | 1±0.9 | 0.68±0.9 | 0.6±0.9 | 0.42±0.9 | 0.23±0.9 | 0.12±0.8 |
| ionosphere scale | SPKDE | 7.5±1 | 7.3±1 | 7.2±1 | 7.1±1 | 7±1 | 7±1 | 7.5±2 |
| | KDE | 7.8±1 | 7.6±1 | 7.5±1 | 7.3±1 | 7.3±1 | 7.3±1 | 7.7±2 |
| | RKDE | 7.6±1 | 7.5±1 | 7.4±1 | 7.4±2 | 7.6±2 | 8.9±4 | 9.9±4 |
| | rejKDE | 7.7±1 | 7.6±1 | 7.4±1 | 7.2±1 | 7.2±1 | 7.2±1 | 7.6±2 |
| ringnorm | SPKDE | -3±0.4 | -8±1 | -10±0.8 | -12±0.8 | -13±0.7 | -13±0.4 | -14±0.4 |
| | KDE | -3±0.4 | -7.8±1 | -9.8±0.8 | -11±0.8 | -12±0.7 | -13±0.4 | -14±0.4 |
| | RKDE | -3.2±0.4 | -8.1±1 | -10±0.8 | -12±0.8 | -13±0.7 | -13±0.4 | -14±0.4 |
| | rejKDE | -3.1±0.4 | -7.9±1 | -9.9±0.8 | -12±0.8 | -12±0.7 | -13±0.4 | -14±0.4 |
| sonar scale | SPKDE | -16±6 | -16±5 | -17±5 | -17±5 | -18±5 | -19±5 | -19±5 |
| | KDE | -16±6 | -16±5 | -17±5 | -17±5 | -18±5 | -19±5 | -19±5 |
| | RKDE | -16±6 | -16±5 | -17±5 | -16±7 | -18±5 | -19±5 | -19±5 |
| | rejKDE | -8.2±9 | -9.4±8 | -9.6±8 | -10±8 | -11±8 | -11±8 | -11±8 |
| splice | SPKDE | 34±0.3 | 34±0.3 | 34±0.3 | 34±0.2 | 34±0.2 | 34±0.2 | 34±0.2 |
| | KDE | 34±0.3 | 34±0.3 | 34±0.3 | 34±0.2 | 34±0.2 | 34±0.2 | 34±0.2 |
| | RKDE | 34±0.3 | 34±0.3 | 34±0.2 | 34±0.2 | 34±0.2 | 34±0.2 | 34±0.2 |
| | rejKDE | 34±0.3 | 34±0.3 | 34±0.3 | 34±0.2 | 34±0.2 | 34±0.2 | 34±0.2 |
| thyroid | SPKDE | -0.86±0.9 | -4.1±0.9 | -5.1±1 | -5.9±0.5 | -6.4±0.4 | -6.7±0.2 | -6.8±0.2 |
| | KDE | -0.89±0.7 | -4±0.7 | -5±0.8 | -5.6±0.4 | -6.1±0.3 | -6.3±0.2 | -6.4±0.2 |
| | RKDE | -0.71±0.9 | -3.9±0.9 | -5±1 | -5.8±0.4 | -6.3±0.3 | -6.6±0.2 | -6.8±0.2 |
| | rejKDE | -0.88±0.8 | -4.1±0.7 | -5.1±0.8 | -5.7±0.4 | -6.1±0.3 | -6.4±0.2 | -6.5±0.2 |
| twonorm | SPKDE | -3.2±0.6 | -3.8±0.5 | -4±0.5 | -4.4±0.4 | -4.6±0.3 | -4.8±0.4 | -5.1±0.4 |
| | KDE | -3.1±0.6 | -3.7±0.5 | -3.9±0.4 | -4.3±0.4 | -4.5±0.3 | -4.7±0.4 | -5±0.5 |
| | RKDE | -3.3±0.6 | -3.9±0.5 | -4.1±0.5 | -4.5±0.4 | -4.7±0.3 | -4.9±0.4 | -5.2±0.5 |
| | rejKDE | -3.2±0.6 | -3.8±0.5 | -4±0.5 | -4.3±0.4 | -4.6±0.3 | -4.8±0.4 | -5.1±0.5 |
| waveform | SPKDE | -7.6±0.3 | -7.7±0.3 | -7.9±0.3 | -8±0.4 | -8.1±0.3 | -8.3±0.3 | -8.3±0.3 |
| | KDE | -7.5±0.3 | -7.7±0.4 | -7.8±0.4 | -8±0.4 | -8.1±0.4 | -8.2±0.4 | -8.3±0.3 |
| | RKDE | -7.6±0.3 | -7.8±0.3 | -8±0.4 | -8.1±0.4 | -8.2±0.4 | -8.4±0.4 | -8.4±0.3 |
| | rejKDE | -7.6±0.3 | -7.8±0.4 | -7.9±0.4 | -8±0.4 | -8.2±0.4 | -8.3±0.4 | -8.4±0.3 |