[Reviews · NeurIPS 2014]

Submitted by Assigned_Reviewer_8

Summary:

This paper considers the problem of robust kernel density estimation, where the authors assume that most observations come from a target density f_tar, but some observations are drawn from a contaminating density f_con. They consider the setting that the observation density f_obs is a known convex combination of f_tar and f_con, but we do not know f_tar or f_con. In practice, the convex combination is not known, but can be treated as a tuning parameter, to adjust between more and less robust estimates. They also have "Assumption A", which states that the contamination density f_con is uniform at r on the support of the target density f_tar, but is less than r otherwise. Under this assumption, they present a decontamination operator that maps a contaminated density to the target density. The decontamination operator rescales the observation density f_obs by a certain amount, then shifts it down, pointwise, to make the resulting function integrate to 1. This addresses the infinite sample, or population version of the problem. To apply this a finite sample, we could first approximate the observation density f_obs by a classic kernel density estimator, and then apply the decontamination operator. However, this does not work in practice, because computing the amount of the shift is not computationally tractable. Instead, they propose taking the L2 projection of the rescaled function onto the convex hull of kernel functions evaluated at each of the sample points. Finding this projection amounts to a quadratic program involving the Gram matrix of L2 inner products of the kernel functions centered at each of the sample points, which can be computed efficiently for many kernels of interest. They eventually prove that the L1 distance between the "SPKDE" estimator and the target density converges to 0 in probability. The authors say that the convergence of their estimator to f_tar, under sufficient conditions, is the first result of its type. In their experiments, they first compare the performance of SPKDE to traditional KDE on artificially generated data from distributions satisfying Assumption 1. SPKDE clearly outperforms KDE. In the experiments on real data, the estimated density based on contaminated training data is compared to an estimated density based on uncontaminated [I assume uncontaminated -- this wasn't entirely explicit] test data. The comparison was done using KL-divergence in both directions. The Wilcoxon signed rank test is used to see if the performance differences are significant across the N datasets.

General Comments:
The paper was quite interesting and readable. I just have a few suggestions. First, density estimation on R is so easy to visualize, I feel like you could do a better job of illustrating the differences between SPKDE, KDE, rejKDE, and RKDE. You could pick synthetic datasets that play to the strengths of each. I think it would be more impactful and interesting.

On the real datasets, using KL divergence in both directions seems reasonable, but is a bit heavy, since you're measuring divergences with respect to another KDE estimate on the test data. [I've assumed that this test data is uncontaminated, but this could be explained more clearly.] What about using test likelihood as another performance measure? This seems very standard for measuring the quality of a distribution's fit to data, and there are no issues of symmetry or having to first "fit" the test data.

With regard to RKDE performing better in one direction of the KL divergence, and the SPKDE performing better in the other direction: can this difference be visualized? Under what assumptions does RKDE perform best, and how do RKDE and SPKDE compare in that situation, and vice versa?

Specific Comments:
- Seems like Proposition 2 is essentially proved by line 136 on page 3. Lemma 1 seems unnecessary to prove Proposition 2. Maybe state Proposition 2 before Lemma 1. And right after say that the transform used "is well-defined more generally" and then give Lemma 1? Or even postpone Lemma 1 until you actually need it in Section 3? As it is currently presented, it's seems like Proposition 2 depends on Lemma 1 (which has a long proof), when really it's quite simple (unless I've misunderstood).

- The acronym SPKDE isn't formally introduced before it's used in the abstract or the introduction.

- Justification for uniformity assumption on f_con: It's unclear what aspects of Assumption A are being supported by the second paragraph on page 3 (starting line 113). Assumption A has two parts: 1) f_con is uniform on the support of f_tar, and 2) The values of f_con outside of supp(f_tar) is bounded above by its value on supp(f_tar). It seems like both of these are important to the method.

- Line 111 on page 3, I think you mean that the union is exactly all densities whose _support_ has finite Lebesgue measure?

- The differences between what's happening in Figure 2 and Figure 3 are pretty subtle. Might be there a more clear illustration of the differences?

- It's rather difficult to find any of these contamination models particularly "natural" or "reasonable". However, on line 182 of page 4, you characterize the assumption in terms of the type of target densities that can be recovered. At that point, it might be worth reminding the reader of the comment on line 111, that any target density with finite support is recoverable under Assumption A your method.
Summary: They present a method for robust kernel density estimation and prove, under certain assumptions on the true density and the contamination, that the L1 distance between their SPKDE estimator and the true density converges to 0 in probability. They claim this result is the first of its kind, and their empirical results are reasonable.

Submitted by Assigned_Reviewer_42

The authors propose a way to obtain a robust Kernel density estimator which can be interpreted as a Kernel density estimator with unequal weights. This estimator arises as a solution to a strictly convex quadratic program. They show some experiments where they compare their method to the usual KDE and to Kim and Scott (2012) robust Kernel density estimator. They use 2 KL divergences and a Wilcoxon test as evaluation metrics and show that their estimator outperforms the competitors ones in one of the KL and in all metrics for the usual KDE.

The paper is well written and structured. Furthermore, they use careful phrasing of what they are doing in most of it. However, my main concern is about their main claim i.e. a consistency result for density estimation (with respect to the uncontaminated density) in a nonparametric setting.
While the author's result is indeed a different consistency result i.e. wrt to f_tar (uncontaminated) rather than f_obs, the assumptions about the set of contamination densities (uniform over the support of f_tar and bounded by a constant outside the support of f_tar) makes me think that their claim is equivalent to showing consistency wrt to f_obs. More precisely, the possible f_tars seem to be just a constant offset of the f_obs. I would like to get this clarified by the authors if possible. Ideally, it would be nice to get consistency rates i.e. that their estimator converges faster than the competitor's but a more detailed explanation of why their assumptions are not too restrictive and how it is not equivalent would be nice.

A few specific comments:

Line 49: What do you mean by kernel weights? This has not been defined yet.

Line 107: "in the Lebesgue sense" it would be more clear to write: almost surely with respect to Lebesgue measure. I guess in this case this statement is equivalent since the density is defined wrt to it.
Line 111: What happens in the unbounded support case? It would be nice to have some comment about this or why it is a hard problem to solve.
Line 136: alpha < = r when eps tends to 0 and alpha > r when eps tends to 1. It would be nice to have some intuition about this cases. Is there a criterion/tradeoff on how to choose it? For example, in the anomaly detection problem you indeed have one (Line 173). In Line 145 you say that it should be chosen so that g is still a pdf but this is not the whole answer since it is still a function of epsilon. Furthermore, in Line 222 you say that this is not computationally tractable.
Line 114: It is not clear in general why the uniformity assumption is natural. I guess you describe why it is the case in the 1 SVM classification but this doesn't imply it is natural. In any case,this assumption on the contamination density sounds a bit restrictive to me.
Line 297: why is this experiments useful? More explanation is required or maybe compare against the RKDE. The idealized setting illustrate that the approximate solution is closer than a naive KDE.
Line 270: This statement should be carefully explained: why does L2 convergence imply L1 convergence in this case? For example,say, in terms of a smoothness condition of the underlying densities.

Quality: this is a well written paper.

Clarity: the importance of the problem is clearly explained and the paper has nice figures to illustrate the concepts.

Originality and Significance: the paper introduces nice ideas however I am still not sure about their main claim.
Summary: The authors propose a method to get a Kernel density estimator with unequal weights which is a solution to a strictly convex quadratic program. Well written and clear motivation but this is not the first consistency result in a nonparametric setting so more detailed explanation about their main claim would be nice.

Submitted by Assigned_Reviewer_43

Robust kernel density estimation

This paper presents a robust version of kernel density estimation. The authors demonstrate the robustness with numerical results and also prove a consistency theorem which states that under some condition the method proposed in the paper can asymptotically recover the original uncontaminated density.
The contamination model is defined as f_{obs}=(1-\epsilon)f_{tar}+\epsilon f_{con}, where f_{obs} is the density of the observed and contaminated data, f_tar is the target density we want to recover and f_{con} is the density of the contaminating noise.
In Proposition 1 the authors prove a somewhat trivial, although very useful lemma which states that if the class of contaminating noise distributions contains more than one different distributions, then exact recovery is impossible.
The main idea of the paper is to assume a special contaminating noise (Assumption A and Figure 1), and then slice this noise off from the bottom of the observed density. The decontaminating transformation is defined in line 144. When we only have finite samples, then this transformation is not known but can be estimated from samples. For this estimation the authors use an RKHS based approach defined in line 230. Similarly to many other RKHS based methods, the proposed algorithm needs to solve an n-dimensional quadratic problem where n denotes the sample size.
The main consistency theorem is stated in Theorem 2.

Questions and Comments:

* Line 232: "which is guaranteed to have a unique minimizer since D^n_sigma is closed and convex". Convexity is not enough to have a unique minimizer (consider e.g. a constant function). We need stronger assumptions, for example strictly convexity.

* Section 4.3: Was there any specific reason to use the KL divergence as a performance metric? Why not L2 distance (which is a metric), or maximum mean discrepancy (which is a metric and easy to estimate), or some other divergences?

* It would be nice to see more figures about the performance of the algorithm similarly to Figure 4. I understand the space constraints make it difficult to place these results in the main article, but the supplementary material could also be used for this purpose. I’m curious about how the algorithm would perform in two and higher dimensions and it would be also nice to see how the performance changes as we vary the number of samples, and change the level of noise contamination.

* It would also be nice to see how the computation time changes as we increase the sample size n. The algorithm has to solve a quadratic problem of dimension n; I assume this is the biggest bottleneck.

Summary: PROS:
The paper is well-written and mostly easy to follow. The authors use proper mathematical notation. The goal of the paper is clear stated, the results are easy to understand.
CONS:
The novelty is somewhat incremental in my opinion, and it is not clear to me how the algorithm would perform in higher dimension and how the computational time scales with the number of instances.

Submitted by Assigned_Reviewer_44

The paper tries to address the problem of robust nonparametric density estimation. Even thought it is an important problem in many applications of statistics and machine learning, as far as I know it has not been understood very well since it is tricky to formulate a good assumption to characterize the contaminating density.

One of the most important contributions of this paper is a simple algorithm with consistency results. The authors first point out that not all contamination models could be recovered. Based on this results, they give their assumption on the contaminating density, which puts a clever restriction on the contaminating model such that the target density is always recoverable. Although assuming the contaminating density depends on the target density seems a little bit strong, the intuition given in the paper is understandable and makes sense. The algorithm SPKDE uses square loss minimization, thus is easy to implemented. One interesting thing is that the algorithm does not include a regularizer term, which is quite normal in kernel related methods. It might be because the minimizer is always restricted insider a probabilistic simplex. This definitely simplifies the algorithm, since kernel density estimation always suffers from the choice of hyper parameter.

The experiments section provides some empirical results on synthetic data and some standard benchmark datasets. The results in the supplementary material are a little bit confusing, because it has negative values for D_{KL}, but no explanation for that is given. The results for D_{KL}(\hat{f} || f_0) and D_{KL}(f_0 || \hat{f} ) looks very different and the authors give some explanation for that. I am curious about which one should be a better metric for this problem. Moreover I suggest the authors to look at some more realistic applications of density estimation and see how their algorithm performs there.
Summary: The paper addresses an interesting and important problem in statistics and machine learning and provides a simple algorithm with clear consistency results, even though the experiments in the paper seem not very convincing.
Author Feedback
Author rebuttal: Response to Assigned_Reviewer_42:

The reviewer asks for clarification as to whether the statement "the possible f_tars seem to be just a constant offset of the f_obs" is correct. The reviewer is precisely correct, we are simply trying to offset f_obs so that only f_tar remains. Figure 1 illustrates the situation we are attempting to decontaminate (Assumption A).

The reviewer's main issue with the paper is that the consistency result for f_tar recovery is not interesting because it seems equivalent to recovery of f_obs and simply applying an offset. We address this approach in lines 222-225. To quote the paper:

"Given a KDE we would ideally like to apply Q_β directly and search over α until max{β \bar{f}_σ^n (·) − α, 0} integrates to 1. Such an estimate requires multidimensional numerical integration and is not computationally tractable. The SPKDE is an alternative approach that always yields a density and manifests the transformed density in its asymptotic limit."
Because this approach is not computationally tractable we apply the scaling and projection technique. Showing that this technique yields the shifted density in its limit is non-trivial.

We will address the rest of Assigned_Reviewer_42's complaints in the order from the author response.

Line 111: Decontamination under Assumption A with f_tar having unbounded support is possible provided that the measure of the support of f_tar is finite. In the situation where the measure of the support of f_tar is not finite, Assumption A would imply that f_con is uniform over a set that has infinite measure, which would further imply that f_con is not a pdf. Thus, when the support of f_tar has infinite measure, Assumption A is violated and we will not recover f_tar by Proposition 3. Whether or not recovering densities with bounded measure is a "hard" problem isn't a question that has a straightforward answer. By Proposition 1 we show that in order for a transform to decontaminate we must make assumptions on f_tar and f_con; the difficulty of decontamination likely depends on these assumptions.

Line 136: Both β and α are functions of ε in such a way that we recover f_tar when Assumption A is satisfied. β goes to infinity as ε goes to one so the amount of density we must remove from the bottom of the scaled density, which is dictated by α, must also go to infinity.

Line 114: The word "natural" was poorly chosen. We simply wanted to mention that the uniformity assumption is consistent with one-class classification approaches.

Line 297: This experiment was included because it depicts how the SPKDE performs in the situation where the data exactly fits our contamination model. From line 299: "To show that the SPKDE’s theoretical properties are manifested in practice we conducted an idealized experiment where the contamination is uniform and the contamination proportion is known." It is also visually represented to give the reader a bit of intuition as to what the SPKDE does in practice.

Line 270: Using Holder's Inequality it is possible to show that, when restricted to a set of finite Lebesgue measure, L^2 convergence implies L^1 convergence. From this observation it is straightforward to show that L^2 convergence of densities implies L^1 convergence of densities. Smoothness conditions are not necessary for convergence.

Response to Assigned_Reviewer_43:

Though we could represent the KDE in a RKHS (provided we use a positive semi-definite kernel), there is no RKHS method in this paper. The method involves "kernels" in L^2 space.

Line 232: While minimizers of convex functions are not necessarily unique in general, finding the closest element in a closed and convex set to another point in a Hilbert space (i.e. projection of a point onto a closed and convex set) always admits a unique minimizer/projection.

Section 4.3: We used the KL divergence metric because it is sensitive to errors on the tail of a density and we felt that it was reasonable to assume that most contamination is probably manifested in the tail. The metrics mentioned by the reviewer are also interesting; we may investigate them in the future.

As for how the algorithm scales with dimension we will simply mention that the Splice and Twonorm datasets are both 60 dimensional.

Assigned_Reviewer_44:

We address the reviewer's comment about the negative KL divergences in line 349, specifically we do not include the C term in these KL divergences. It would probably be worthwhile to mention the negative values more explicitly.

Assigned_Reviewer_8:

Thank you for you comments. We plan on making the changes you suggested.